# Assessment and Management of HIV-Associated Cognitive Impairment: Experience from a Multidisciplinary Memory Service for People Living with HIV

**DOI:** 10.3390/brainsci9020037

**Published:** 2019-02-08

**Authors:** Kate Alford, Sube Banerjee, Eileen Nixon, Clara O’Brien, Olivia Pounds, Andrew Butler, Claire Elphick, Phillip Henshaw, Stuart Anderson, Jaime H. Vera

**Affiliations:** 1Department of Global Health and Infection, Brighton and Sussex Medical School, Brighton BN1 9PX, UK; K.Alford@bsms.ac.uk; 2Centre for Dementia Studies, Brighton and Sussex Medical School, Brighton BN1 9PX, UK; S.Banerjee@bsms.ac.uk; 3Sussex Partnership NHS Foundation Trust, Worthing BN13 3EP, UK; Claire.Elphick@sussexpartnership.nhs.uk (C.E.); Phillip.henshaw@sussexpartnership.nhs.uk (P.H.); 4Brighton and Sussex University Hospitals NHS Trust, Brighton BN2 1ES, UK; E.Nixon@bsuh.nhs.uk (E.N.); Clara.OBrien@bsuh.nhs.uk (C.O.); Olivia.Pounds@bsuh.nhs.uk (O.P.); Stuart.Anderson@bsuh.nhs.uk (S.A.); 5Department of Medicine, Brighton and Sussex Medical School, Brighton BN1 9PX, UK; Andrew.Butler@bsms.ac.uk

**Keywords:** HIV, HIV-associated neurocognitive disorder, cognitive impairment, patient care

## Abstract

As the HIV population ages, the prevalence of cognitive impairment (CI) is increasing, yet few services exist for the assessment and management of these individuals. Here we provide an initial description of a memory assessment service for people living with HIV and present data from a service evaluation undertaken in the clinic. We conducted an evaluation of the first 52 patients seen by the clinic. We present patient demographic data, assessment outcomes, diagnoses given and interventions delivered to those seen in the clinic. 41 patients (79%) of those seen in the clinic had objective CI: 16 (31%) met criteria for HIV-associated Neurocognitive Disorder (HAND), 2 (4%) were diagnosed with dementia, 14 (27%) showed CI associated with mental illness and/or drugs/alcohol, 7 (13%) had CI which was attributed to factors other than HIV and in 2 (4%) patients the cause remains unclear. 32 (62%) patients showed some abnormality on Magnetic Resonance Imaging (MRI) brain scans. Patients attending the clinic performed significantly worse than normative scores on all tests of global cognition and executive function. Interventions offered to patients included combination antiretroviral therapy modification, signposting to other services, case management, further health investigations and in-clinic advice. Our experience suggests that the need exists for specialist HIV memory services and that such a model of working can be successfully implemented into HIV patient care. Further work is needed on referral criteria and pathways. Diagnostic processes and treatment offered needs to consider and address the multifactorial aetiology of CI in HIV and this is essential for effective assessment and management.

## 1. Introduction

In the UK over 100,000 people are chronically infected with HIV, with over 6000 new cases reported each year [1]. Cognitive impairment (CI) in patients living with HIV (PLWH) is commonly reported and is likely to become an increasingly important issue as this population ages. HIV-associated dementia, which occurred in up to 50% of PLWH prior to the introduction of potent combination antiretroviral therapy (cART), is now rare, however, mild to moderate CI is widely reported and results in lower quality of life, poorer adherence to medication, increased unemployment and reduced life expectancy [2,3,4]. Prevalence rates vary widely, depending on definitions of CI, with estimates reported as high as 52% [5]. Studies with tighter definitions estimate that between 14–28% of PLWH over 50 years old have CI [6], which may or may not be associated with HIV. The term HIV-Associated Neurocognitive Disorder (HAND) has been coined to refer to a spectrum of impairments (Asymptomatic Neurocognitive Impairment (ANI); Mild Neurocognitive Disorder (MND); and HIV-associated Dementia (HAD)). The main method used to define HAND are the Frascati criteria [7]. These involve neuropsychological testing across multiple cognitive domains, HAND is diagnosed when two or more domains are more than one standard deviation below normative scores and for MND and HAD when daily functions are affected. Recent studies have suggested that the Frascati criteria have poor sensitivity and specificity, and do not take into account the complexity of pathogenic mechanisms likely to contribute to CI in PLWH [7,8,9]. This lack of clarity in definition and diagnosis is a significant issue which affects the management of PLWH with CI. Indeed, the pathogenic mechanisms causing CI are often multifactorial, including complex immunopathological processes controlled by HIV factors, the direct effects of cART, and host factors (e.g., co-infections, cardiovascular and cerebrovascular diseases, psychiatric illnesses, the effect of age and age-related illnesses on the brain and lifestyle factors, including social isolation [10]).

Given the aetiological complexities PLWH with CI experience, where causation, diagnosis and/or management may all be unclear, patients might benefit from a multidisciplinary response. Serial consultations with HIV physicians, neurologists, dementia specialists, neuropsychologists and clinical psychologists may be time-consuming, uncoordinated and imperfectly synthesised. Additionally, in terms of management they may require psychological input and sign-posting to other specialist services for further investigation or management. While there are criteria to identify HAND, few recommendations exist on how best to manage and follow-up patients found to have CI and treated HIV. No guidance exists on how to address the clinical, social and psychological needs of HIV patients with progressive cognitive impairments and early dementia, regardless of CI pathology. In practice, the patient will often be referred to multiple services adding complexity, reducing efficiency and increasing social and economic burden. Moreover, in clinical experience we find joint clinics run outside of the HIV clinical setting have higher levels of non-attendance than equivocal clinics held within our normal HIV service setting. Importantly, patients complaining of CI and sign-posted to other services for assessment or investigation often struggle to remember where to go or who they are seeing and report feeling anxious about disclosing HIV-status, creating a further barrier to care [11].

In order to address these issues and generate a clear and efficient pathway for PLWH with CI and given the large numbers of people growing older with HIV in the Brighton area, the Orange Clinic was set up in 2016 as a collaboration between HIV services, memory assessment services and Brighton and Sussex Medical School. It aims to provide a single point of call for expert, multidisciplinary assessment, management and advice on care for PLWH with concerns about CI. Here we provide an initial description of the service and present data from the service evaluation undertaken of the clinic as a case study. We then use these experiences to make preliminary recommendations for the assessment and management of PLWH with cognitive complaints. 

## 2. Materials and Methods

### 2.1. Clinical Setting

The Orange Clinic team consists of (i) an HIV consultant physician, (ii) a consultant old age psychiatrist skilled in dementia assessment and management, (iii) a neuropsychologist with a psychology assistant, (iv) a clinical psychologist, (v) a HIV clinical nurse consultant and (vi) virtual support from neurology and neuroimaging services. The clinic operates for assessments for two sessions on a monthly basis with an additional monthly follow-up session. In one day, the clinic aims to have completed multiple assessments and to triangulate the information from these with history and investigations to generate for each referral a full assessment and management plan. Referrals to the clinic may come from any healthcare professional involved in patient care and indications for referral are PLWH with cognitive complaints, or unexplained cognitive disorders where the referring team would like help with diagnosis and/or management. Prior to clinic attendance, a multidisciplinary virtual (without a patient present) case-based discussion of each patient is organised, this includes: a background review, evaluation of current knowledge/assessments and a review of the need to request further investigations (e.g., MRI or Lumbar Puncture (LP)) prior to assessment. Relatives or friends of the patients are encouraged to attend the clinic assessment, to provide a collateral history. The patient is interviewed jointly by the consultants in HIV, psychiatry and clinical psychology. The patient has a detailed neuropsychological assessment completed and all these data and any diagnosis are discussed by the multidisciplinary team as a whole and with the PLWH, following which a formulation and management plan is agreed, co-produced by all. 

### 2.2. Neuropsychological Assessments

We have developed a test battery that is adapted to the clinical needs of the clinic. Using established assessments, proven sufficiently valid and reliable across a variety of populations, each patient attending the clinic is assessed in the broad domains of premorbid IQ and global cognition across multiple cognitive domains, including memory, attention, language processing, visuospatial processing and executive functioning (see Table 1 for neuropsychological tests performed).

### 2.3. Psychological Assessment

Depression Anxiety and Stress Scale—21 item version (DASS21): The DASS21 is a short form version of the Depression Anxiety and Stress Scale [23]. Scores on each domain are categorised into normal (Depression, 0–9; Anxiety, 0–7; Stress, 0–14), mild (Depression, 10–13; Anxiety, 8–9; Stress, 15–18), moderate (Depression, 14–20; Anxiety 10–14); Stress 19–25), severe (Depression, 21–27; Anxiety, 15–19; Stress, 26–33) and extremely severe (depression 28+; Anxiety, 20+; Stress 34+). 

### 2.4. Quality of Life Assessments

EuroQol five-dimension descriptive system (EQ-5D-5L): The EQ-5D-5L [24] is a brief self-reported measure of generic health that consists of five dimensions. There is extensive literature supporting the validity and reliability of the measure across many conditions and populations [25,26]. In the clinic, to provide a brief sense of overall health perceptions, we recorded how patients rated their health out of 100, with 0 being the worst health imaginable. 

DEMQOL: DEMQOL [27] is a dementia-specific interviewer-administered health related quality of life questionnaire appropriate for use at all stages of dementia severity with self- (28 items) and proxy-report (32 items) versions. It has been used in memory service populations with all levels of CI, including those with Mild Cognitive Impairment and none [28].

### 2.5. Diagnosis

Criteria for specific diagnoses within the clinic are based on interpretation of an individuals’ test scores, along with the clinical assessment of other relevant factors. These include: patients’ mood (both self-reported and on mental state examination); neuropsychological presentation, including the collateral history (i.e., a profile may suggest co-existing Alzheimer’s disease (AD)); past medical history and imaging results (e.g., If traumatic brain injury has occurred in the past, are the pattern of neuropsychological test results more consistent with damage caused to this area by the injury); effort; fatigue; test properties, including the relative psychometric properties of individual tests and the commonality of abnormally low scores within the general population; and limitations on test performance due to motor or sensory deficits and language barriers (e.g., English not being a patient’s first language). Thus, in the clinic, multiple variables, not just scaled test scores, are considered when assessing the presence and aetiology of impairment and diagnosis.

## 3. Results

From June 2016 to May 2018 the Orange Clinic assessed 52 patients. Demographic, HIV and health data of patients attending the clinic are presented in Table 2.

Of the 52 patients seen, 42 had HAND based on Frascati criteria. Using our diagnostic criteria, we found 11 (21%) patients to have no objective CI, 16 (31%) were diagnosed with HAND, 2 (4%) were diagnosed with dementia (one AD, 1 unspecified). In 14 (27%) the CI was assessed as being secondary to mental health difficulties (depression, anxiety, OCD, poor sleep) and/or drug/alcohol abuse, 7 (14%) had CI which was attributable to factors other than HIV (2 prior Traumatic Brain Injury (TBI), 4 cerebrovascular diseases and 1 cortical dysplasia), and in 2 (4%) cases the CI cause remains unclear (both are recent cases and we are awaiting LP/MRI or other investigations). 

All but one patient had a new (*n* = 28) or recent MRI (*n* = 23) and 32 (62%) had some type of MRI brain abnormality. White matter (WM) hyperintensities were the most common finding (*n* = 22), of these 11 were considered vascular and indicative of cerebrovascular issues (i.e., ischaemia or small vessel disease). 8 scans showed evidence of cortical atrophy and 6 of subcortical atrophy. 4 patients demonstrated abnormalities consistent with HIV-associated CI (e.g., damage from previous HIV-associated encephalitis or leukoencephalopathy). 5 patients’ scans had ‘other’ abnormalities, including meningioma, pontine lesions, left insular lesions, hyperintensities of the pons and siderosis. 

Lumbar Punctures (LPs) were requested for 26 patients attending the clinic. Patients did not receive a LP when it was considered unnecessary (e.g., likely mental health causation for CI) or when patients refused it. Within our cohort there was no recorded cases of CSF escape. 

The majority of patients were assessed using the Montreal Cognitive Assessment (MoCA) [12] prior to clinic attendance and the mean score was 23.6 (2.6). All other neuropsychological test scores were converted into Standard Scores (SS) to allow comparison of data (Mean = 100, SD = 15). Test of premorbid functioning (TOPF) clinic mean was 99.2 (11.8) and Reynolds Intellectual Screening Test (RIST) clinic mean was 99.1 [15], indicating that the general intelligence of our sample was similar to the population mean. Mean performance on Repeatable Battery for the Assessment of Neuropsychological Status (RBANS) total scale was 80 (17.3) and on each RBANS domain: immediate memory, 82.2 (20.8); visuospatial, 94.5 (18.9); language, 86.5 (15.5); attention, 79.9 (16.9); delayed memory, 79.2 (21.5). Figure 1 shows the mean RBANS scores seen in those attending the clinic against average scores reported in the literature from patients with Alzheimer’s dementia and mild cognitive impairment [29,30]. Interestingly, women (*n* = 11) attending the clinic were outperformed by males on all measures, with statistically significant differences on measures of premorbid functioning (males in our cohort had higher premorbid IQ) (*p* < 0.01) as well as the mazes task (*p* = 0.02), visuospatial (*p* < 0.01), language (*p* = 0.001) and attention (*p* = 0.02) indices of the RBANS and the RBANS overall score (*p* = 0.01). Differences seen are likely due to differences in premorbid IQ, and that women attending the clinic tended to be from different cultural (i.e., where English is not the first language) and socio-economic backgrounds to the men. 

One-sample *t*-tests suggest that scores on Total RBANS score were significantly worse in patients attending our clinic (80 ± 17.32) than normative scores (100 ± 15), *t* (51) = −8.31, *p* < 0.001. Indeed, our clinical cohort performed on average significantly worse than normative scores across all RBANS subdomains; Immediate memory (82.19 ± 20.75), *t* (51) = −6.19, *p* < 0.001; Visuospatial (94.54 ± 18.85), *t* (51) = −2.09, *p* = 0.04; Language (86.46 ± 15.46), *t* (51) = −6.31, *p* < 0.001; Attention (79.87 ± 16.9), *t* (51) = −8.59, *p* < 0.001; Delayed memory (79.21 ± 21.55), *t* (51) = −6.96, *p* < 0.001). Patients also completed a range of executive functioning tests. One-sample *t*-tests again revealed those attending the clinic performed significantly worse than normative scores (100 ± 15) in all executive functioning tests: NAB screening module mazes (89.63 ± 18.66), *t* (50) = −3.97, *p* < 0.001; DKEFS colour word interference test (83.79 ± 18.72), *t* (41) = −5.61, *p* < 0.001; Trails Test A (81.40 ± 20.14), *t* (49) = −6.53, *p* < 0.001; Trails Test B (74.37 ± 23.07), *t* (42) = −7.29, *p* < 0.001.

### 3.1. Mental Health and Quality of Life Assessments

The mean score on total DASS21 across the patients attending the clinic was 27 (SD = 13.4), the mean depression score was 8 (52), the mean anxiety score was 8 (4.9) and the mean stress score 11.0 (5.3). Across the clinic the mean ED-5Q-5L score was 65 (23.9) and mean DEMQOL total score was 70 (17.1). We found no correlation between EQ-5D-5L score and DEMQOL total score and no correlation between any of the HIV data variables and either DEMQOL total score or ED-5Q-5L score. The scores on DEMQOL are indicative of a lower quality of life than in the population of those attending memory assessment services where the mean DEMQOL score at assessment of 197 attenders was 89.6 (14.1); *t* (247) = 8.47, *p* < 0.0001 [31]. 

### 3.2. Management

Patients attending the clinic received a variety of interventions, referrals and advice. These are summarised in Table 3.

Of the 52 patients who attended the clinic, 23 are now discharged. Of those discharged, 9 had no objective CI, 8 had CI due to mental health issues which were best served by other services, 1 patient had CI due to prior HIV encephalitis, 1 had CI due to a previous traumatic brain injury, and 3 had mild HAND. 10 of these patients were discharged following initial appointment. These patients were given in-clinic advice (*n* = 9), referred to mental health services (*n* = 3), referred to community drug support services (*n* = 1) and provided with a recommendation for strict control of cardiovascular risk factors (*n* = 1). The other 12 were discharged following satisfactory feedback from further investigations or once a management plan had been fully implemented and the one patient with Alzheimer’s disease died.

29 patients are still open to the clinic. Of these 15 are due to have repeat cognitive testing to assess progression and/or impact of management plan and a further 8 are due to be followed-up following implementation of management plans (e.g., psychological therapy, in-clinic advice on lifestyle/mental health management e.g., sleep hygiene, medication changes, stricter control of CVD risk factors). 2 patients will be reviewed again following MRI/LP/other health assessment feedback before a management plan is devised. Finally, 4 patients remain open to the clinic but are not due to be seen unless needed, these patients are all either responding well to their current plan or have been referred to another health service. 

## 4. Discussion

Historically, PLWH have looked to their HIV clinic to manage their medical, and in some cases, social and psychological needs. HIV services have always been active in adapting and innovating care models to provide for the changing needs of their patients. The Orange Clinic represents such a model—a novel, needs-driven, efficient and coordinated service for the ageing population of PLWH who experience neurocognitive issues, which is in keeping with NHS England’s 5-year forward view. 

Patients attending the clinic varied widely in age (36–84 years), and in time since virus acquisition (1–34 years). Our cohort is distinctive in that 73% of patients were MSM, whereas current UK statistics show 55% of PLWH identify as heterosexual [32]. All but one patient was on cART, which is in line with the national average of 96%, and all but four patients had a VL <40 copies/mL. Of these, three were diagnosed with HAND and one with dementia, all had their cART switched (or in one case re-started). 

Based on Frascati criteria, 81% of patients attending the clinic would be diagnosed with HAND. Using our diagnostic processes, we found 79% had an objective CI, however of these, 61% had a reason for this that was not HIV-related and only 39% had ‘true’ HAND. This highlights that the Frascati criteria when applied to data in isolation from other factors (e.g., patient mood, nature of neuropsychological presentation, past medical history, imaging data, effort, fatigue, relative psychometric properties of individual tests) do not correlate well with interpretation of test scores by experienced neuropsychologists, largely because this model does not account for the extensive range of possible confounding variables that are present in clinical samples. Whilst these may be of use in research settings, in clinical practice the use of the Frascati model may miss much that is of clinical importance and lacks utility when considering and producing management plans. 

This high prevalence of CI detected is in line with the purpose of the clinic, and the heterogeneity of causation of CI in HIV is well illustrated. None had HIV dementia, our main test of global functioning (RBANS) supports research indicating the pattern of CI from pre and post cART eras has changed. AIDS-dementia complex, seen frequently in the pre-cART era, was characterized by progressive subcortical dementia with prominent degeneration of cognitive and motor functions. Our clinic patients diagnosed with HAND (based on our diagnostic criteria) showed impairments in tests of immediate and delayed memory, with mild impairments in attention, visuospatial skills and language—demonstrating a more subtle subcortical involvement and cortical involvement possibly influenced by age [33]. 

It is significant that 27% of those who attended the clinic had a mental health condition which was likely to be responsible for their subjective and objective CI. A number of other patients attending the clinic were also experiencing poor mental health which was not thought to be causing their CI. It is also striking that 44% of patients were taking antidepressant medication. Current management of depression in HIV relies primarily on antidepressant medication [34] with open-label trials of different antidepressants, across HIV illness stage, showing response rates of 70–90% (equivalent to non-HIV infected populations) and good levels of tolerance [35,36,37]. A recent systematic review examining the efficacy of different interventions for PLWH experiencing depression, found that psychotropic and HIV-specific psychological interventions incorporating a cognitive behavioural component were most effective [38]. In our clinic it has been vital to be able to address the mental health issues of patients. This has involved either referrals to HIV specialist mental health services (if mental health issues are related to HIV issues such as diagnosis) (*n* = 8) or to general mental health services (*n* = 16). 

Where social isolation or drug and alcohol issues are a concern, patients have been directed to different community services (*n* = 6). We have also given in-clinic advice on mental health management (*n* = 7) and initiated treatment ourselves when urgent. Our data show how important it is to have the skills available to assess mental health and clear pathways for those attending HIV services who have problems with CI. The relationship between HIV, mental illness, and CI is complex. The psychological and social impact of an HIV diagnosis (stigma, discrimination and isolation) can contribute to symptoms of depression and anxiety, and therefore to subjective and objective CI. In addition, HIV replication in the Central nervous system (CNS) causes depression via the modification of brain structures [39], somatostatin dysregulation [40] and increased inflammatory cytokines [41], as well as lowering cognitive reserve and impairing cognitive function directly and via co-infection. Added into this are lifestyle factors such as drug and alcohol use which may be exacerbated by mental health issues and will have a negative impact on cognitive function. 

The finding that our clinic population has a much lower quality of life than that of people attending memory services is of interest. This suggests that these concerns about CI in the context of HIV, and with high comorbidity with mental disorders, act to decrease the quality of life of those with HIV and CI. It is striking that the condition-specific measure DEMQOL picked this up while the generic EQ-5D-5L did not. We used DEMQOL, despite it having been designed for individuals with dementia and MCI, as it captures many of the experiences which likely dictate quality of life for those with HIV and CI such as feeling understood when trying to express oneself and having enough social company. In the clinic while the results need to be interpreted with caution, it does provide us with useful extra information and informs interventions offered. 

Early (and efficient) detection of milder forms of HAND (i.e., ANI) is a priority, with the British HIV Association (BHIVA) recommending all PLWH should be screened annually for CI. However, there are barriers to this in terms of the lack of quick, efficient and valid screening tools (particularly with the ability to detect milder forms of HAND) and to the lack of evidenced-based treatment guidelines about what to do once it is found. The clinical relevance of seeking to ‘diagnose’ ‘asymptomatic’ CI has been called into question as wasteful of resources and needlessly worrying to patients. Recent findings have shown that ANI is predictive of further cognitive decline [42] but the main moderators of these changes are common to all, including older age, cardiovascular risk factors (diabetes, hypertension, smoking), presence of depression and other health comorbidities [42,43,44]. Medication treatment strategies for CI have generally proved disappointing, within the clinic we modified cART regimens only due to patient complaint of side effects (and would have modified if presence of CSF HIV RNA was detected); besides this little recommendation or indeed clarity of evidence exists. Improvements in neurocognition using medications with better CNS penetration or maraviroc is widely debated [45,46,47] and while evidence points to neurotoxicity associated with some regimens (i.e., Efavirenz), clarity around the most efficacious cART regimen to use with patients living with HAND is unclear. In HIV-negative populations, potential benefits of early recognition of CI include protection from unsafe situations, increased quality of life through information sharing, and improved life planning [47]. These are also likely to be relevant to those with HIV. 

Interventions addressing cardiovascular risk factors and mental wellbeing appear to be of value [42,44]. At the clinic we offer patients informal advice on lifestyle (e.g., diet, exercise, sleep hygiene, drugs/alcohol intake), mental health management, adherence to cART and cognitive strategies, along with recommendations to their GP on the control of CVD risk factors. We plan quantitative evaluation of the effectiveness/utility of the clinic, along with qualitative experiential data from current patients. Furthermore, we are planning to evaluate the clinical use of inflammatory biomarkers and markers associated with HAND to aid diagnostic uncertainty in PLWH. A biomarker that has shown promise is the neurofilament light chain protein (NFL) [48] which could be used in routine clinical practice to define those patients more likely to have HAND. APOE ε4 carriage has also been suggested as a potential predictive marker for HAND. Evidence regarding the role of APOE in PLWH with HAND is conflicting. A study based on the Hawaii Aging with HIV cohort [49] (*n* = 182) associated APOE ε4 carriage with HAND, but only with older participants. A large ethnically diverse cohort (*n* = 2399) found an association with APOE ε4 and acceleration of HIV disease, but not HAD (the authors only examined HAD and did not assess complete HAND spectrum) [50]. More recently in the CHARTER study (*N* = 466) participants received a comprehensive HAND assessment and no association was found between APOE ε4 carriage and HAND [51]. Interestingly, the authors note that differences between studies may be due to the age of the sample; only 4% of their sample was over 60 years of age, compared to 25% in the Hawaii cohort and therefore results may not preclude the emergence of an association between HAND and APOE ε4 status as this population ages. 

## 5. Conclusions

In the UK, 48% of those accessing HIV services are now aged 45 and over, with large increases seen in those of an older age in the past decade. Recent modelling work predicts that by 2030 73% of PLWH will be over 50 years of age [52]. Multidisciplinary working is vital to the successful management of such patients where complex multimorbidity is likely to be the norm. This service evaluation provides tentative evidence that the need exists, that the model of care we have developed is feasible and that there may be value in establishing similar models of working in HIV care for those with impairment in cognitive function. 

## Figures and Tables

**Figure 1 brainsci-09-00037-f001:**
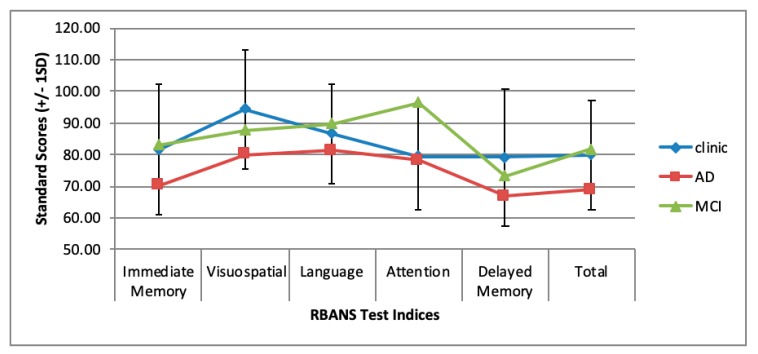
Average RBANS scores of clinic patients against average scores reported in the literature from patients with Alzheimer’s dementia and mild cognitive impairment [29,30].

**Table 1 brainsci-09-00037-t001:** Neuropsychological tests used in the Orange Clinic.

Neuropsychological Test	Test Description
Montreal Cognitive Assessment (MoCA) [12]	Completed prior to clinic attendance, this screening test is designed to detect Mild Cognitive Impairment (MCI) and Alzheimer’s disease across five domains. Scored out of 30, ≥26 is considered normal, ≤23 indicative of MCI and >17 of AD [12].
Test of premorbid functioning UK version (TOPF) [13]	The TOPF is a word reading task designed to assess estimated premorbid functioning [13].
Reynolds Intellectual Screening Test (RIST) [14]	The RIST provides a measure of general intelligence (14). An index score is calculated based on its two subtests which examine vocabulary knowledge and nonverbal reasoning.
Repeatable Battery for the Assessment of Neuropsychological Status (RBANS) [15,16]	The RBANS is a brief assessment battery from which five index scores can be derived from each of its subtests: memory, language, visuospatial processing and attention. It has demonstrated strong diagnostic accuracy for Alzheimer’s disease and MCI [17,18].
Trail Making Test [19]	The Tombaugh (2004) version of the Trail Making Test was employed as a measure of executive functioning [19], with Part B noted to measure cognitive flexibility along with set shifting [20].
Delis Kaplan Executive Function Scale (DKEFS) colour word interference test [21]	This subtest from the DKEFS [21] is a version of the classic ‘stroop’ colour-word interference task, which was employed as a test of executive functioning to measure inhibitory control.
Neuropsychological Assessment Battery—Screening Module (NAB-SM) mazes task [22]	One subtest from the screening module of the NAB-SM [22] is included in this battery to provide a basic test of planning.

**Table 2 brainsci-09-00037-t002:** Demographic, HIV and health data of patients attending the clinic.

Variable	Total
**Demographic Data**	
Median age in years (range) **	55 (36–84)
Male (%)	41 (79)
White (%)	43 (83)
Black British/African (%) Asian MSM (%) Heterosexual (%) Other (%)	7 (13) 2 (4) 38 (73) 13 (25) 1 (2)
Referral source:	
HIV Physician (%)	33 (63)
Memory Assessment Service (%)	4 (8)
Neurology Service (%)	3 (6)
Community HIV specialists (%)	3 (6)
Mental Health Services (%)	1 (2)
Other (%)	8 (15)
**HIV Clinical Data**	
Time since HIV diagnosis (years) **	17 (1–34)
Duration of cART (years) **	13 (1–22)
Nadir CD4 count (cells/μL) *	312.27 (207.48)
Current CD4 count (cells/μL) *	689.37 (279.45)
Current CD8 count (cells/μL) *	920.98 (410.06)
CD4:CD8 ratio	0.87 (.47)
VL >40 (%)	4 (8)
On cART (%)	51 (98)
On PI based regimen (%)	12 (24)
On PI based and NRTIs regimen (%)	13 (25)
On NRTI and NNRTI regimen (%)	18 (35)
On another combination of regiment (%)	9 (17)
**Health Data**	
Smoking (% smokers)	12 (23)
Alcohol intake (units/week) **	2 (0–90)
Recreational drugs (% use) Cannabis (%) Methedrone GHB Other	16 (31) 10 (16) 4 (25) 3 (19) 5 (31)
CVD Risk (RISK2) *	10.34 (9.45)
Mean number (range) non-HIV medications Antidepressants (%)	4.6 (0–20) 24 (46)
Statins (%)	14 (27)
Antiplatelets (%)	8 (15)
Analgesia (%)	12 (23)
PPI (%)	12 (23)
Other (%)	33 (63)
Polypharmacy (≥3 non-HIV medications, %)	26 (50)

MSM, men who have sex with men; cART, combination antiretroviral therapy; PI, Protease Inhibitor; NRTI, Nucleoside reverse transcriptase inhibitor; NNRTI, Non-nucleoside transcriptase inhibitor; CVD, cardiovascular disease; PPI, proton pump inhibitor. All values are expressed as *n*, unless otherwise stated. * mean (standard deviation). ** median (range).

**Table 3 brainsci-09-00037-t003:** N and type of intervention administered by the clinic.

Management Type	*N*
MRI	20
LP	22
Request Blood Tests	6
CT/ PET Scan	2
Neurology advice	5
Request genotyping of CSF virus	2
Switch/review/intensify cART	14
Request other health investigation	9
Review/modify co-medication	11
Refer to generic Mental Health Services	16
Refer to HIV Mental Health Specialist Services	8
Refer to community HIV Specialist nursing team	12
Refer to Other Health Services	5
Signpost to a non-HIV community service	6
Control CVD risk factors	11
In clinic advice: Cognitive strategies/remediation	10
In clinic advice: Lifestyle	14
In clinic advice: Mental health management	7
In clinic advice: cART adherence	5
Total	185

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
