# Peer review of "Assessment and Management of HIV-Associated Cognitive Impairment: Experience from a Multidisciplinary Memory Service for People Living with HIV"

_brainsci, 2019, doi:10.3390/brainsci9020037_

Reviewer 1 Report

This is an interesting manuscript and informative for other services to understand this model of care.

Out of interest how many women were in the study?  Did you test for any differences in gender regarding CI?

Line 95 : could you clarify what you exactly mean by "virtual" as an international audience may have differing interpretations of virtual.

Line 227 you allude to "Table 4". There is no table 4 in the manuscript. ? typo?

Line starting 302-  I am not aware if this journal has a "conclusion" heading but I felt when I finished reading  I had to re- read from 302 as it seemed like an abrupt ending. A conclusion heading may help this.

Author Response

Responses to reviewer 1 comments

This is an interesting manuscript and informative for other services to understand this model of care.

Point 1:

Out of interest how many women were in the study?  Did you test for any differences in gender regarding CI?

Included in results (line 170): 

‘Interestingly, women (n =11) attending the clinic were outperformed by males on all measures, with statistically significant differences on measures of premorbid functioning (males in our cohort had higher premorbid IQ) (p<0.01) as well as the mazes task (= 0.02), visuospatial (p<0.01), language (= 0.001) and attention (p= 0.02) indices of the RBANS and the RBANS overall score (= 0.01). Differences seen are likely due to differences in premorbid IQ, and that women attending the clinic have tended to be from different cultural (i.e. where English is not the first language) and socio-economic backgrounds to the men’.  

Line 95: could you clarify what you exactly mean by "virtual" as an international audience may have differing interpretations of virtual.

‘Prior to clinic attendance, a multidisciplinary virtual (without patient present) case-based discussion of each patient is organised’

Line 227 you allude to "Table 4". There is no table 4 in the manuscript. ? typo?

Deleted. There was a Table 4 which we decided not to submit.

Line starting 302-  I am not aware if this journal has a "conclusion" heading but I felt when I finished reading  I had to re- read from 302 as it seemed like an abrupt ending. A conclusion heading may help this.

Conclusion included from line 311

Reviewer 2 Report

This is a sophisticated report on experience with a comprehensive HIV clinic tailored to deal with the increasing number of aging HIV infected individuals whose infection is controlled, yet will have a higher incidence of cognitive impairment and mood disorders compared to uninfected persons.  The authors present a comprehensive understanding complicated and divisive Neuro-AIDS field and an exhaustive description of the tests employed.  The establishment of such comprehensive HIV treatment centers should be the way of the future, for all of the reasons also well described.  This reviewer’s critiques are relatively minor. 

First, the report is somewhat lacking in enthusiasm and interest in future directions.  Possibly this is due to numerous authors or the tendency of Global Heath personnel to have an observational outlook rather than interventional goals, but it leaves the reader wondering where the authors are going (scientifically), why they wrote the report (other than to continue their funding), and whether they might score as depressed.  

Most of us in the Neuro-AIDS field are aware that once virus is effectively suppressed in CSF and periphery, the remaining inflammatory and medication effects on cognition and mood are much more subtle.  Many may overlap with normal aging, but that is not a reason not to persevere with understanding mechanisms and treating consequences.  Much has been learned about the Human immune system from the study of HIV and much may be learned about general aging and cognition from the HIV cohorts as well.  

Although this report mentions inflammation as a mediator of impairment and mood disorders, there is no measurement of inflammation and no suggestion of (alternative, dietary, physical or pharmacologic) treatment which might reduce inflammation.

Another thread which is missing from this report involves genetics.  For instance, there are strong correlations with neurocognitive impairment and the APOE4 allele.

Author Response

Reponses to comments provided by reviewer 2

Point 1:

First, the report is somewhat lacking in enthusiasm and interest in future directions.  Possibly this is due to numerous authors or the tendency of Global Heath personnel to have an observational outlook rather than interventional goals, but it leaves the reader wondering where the authors are going (scientifically), why they wrote the report (other than to continue their funding), and whether they might score as depressed.  

The manuscript was intended to present our experience of managing people with HIV with cognitive difficulties in a memory clinic set up only for people with HIV. Scientifically we hope the manuscript will encourage researchers to generate research questions on how to develop and evaluate models of care for people with HIV with cognitive impairment. For instance, what referral criterium should be used? What screening tool? What staff needed to deliver such as service? And what would be the best outcome to define success? More importantly, how patients with cognitive difficulties want their care to be provided? 

Most of us in the Neuro-AIDS field are aware that once virus is effectively suppressed in CSF and periphery, the remaining inflammatory and medication effects on cognition and mood are much more subtle.  Many may overlap with normal aging, but that is not a reason not to persevere with understanding mechanisms and treating consequences.  Much has been learned about the Human immune system from the study of HIV and much may be learned about general aging and cognition from the HIV cohorts as well.  

Point 2:

Although this report mentions inflammation as a mediator of impairment and mood disorders, there is no measurement of inflammation and no suggestion of (alternative, dietary, physical or pharmacologic) treatment which might reduce inflammation.

Line 309: ‘Furthermore, we are planning to evaluate the clinical use of inflammatory biomarkers and markers associated with HAND to aid diagnostic uncertainty in people with HIV. A biomarker that has shown promise is the neurofilament light chain protein (NFL) (48) which could be use in routine clinical practice to define those patients more likely to have HAND’.  

Point 3:

Another thread which is missing from this report involves genetics.  For instance, there are strong correlations with neurocognitive impairment and the APOE4 allele.

Line 313 ‘APOE e4 carriage has also been suggested as a potential predictive marker for HAND. Evidence regarding the role of APOE in PLWH with HAND is conflicting. A study based on the Hawaii Aging with HIV cohort (49)(n=182) associated APOE e4 carriage with HAND, but only with older participants. A large ethnically diverse cohort (n=2399) found an association with APOE e4 and acceleration of HIV disease, but not HAD (the authors only examined HAD and did not assess complete HAND spectrum) (50). More recently in the CHARTER study (N=466) participants received a comprehensive HAND assessment and no association was found between APOE e4 carriage and HAND (51). Interestingly, the authors note that differences between studies may be due to the age of the sample; only 4% of their sample was over 60 years of age, compared to 25% in the Hawaii cohort and therefore results may not preclude the emergence of an association between HAND and APOE e4 status as this population ages.’